# Automatic Identification of Ultrasound Images of the Tibial Nerve in Different Ankle Positions Using Deep Learning

**DOI:** 10.3390/s23104855

**Published:** 2023-05-18

**Authors:** Kengo Kawanishi, Akihiro Kakimoto, Keisuke Anegawa, Masahiro Tsutsumi, Isao Yamaguchi, Shintarou Kudo

**Affiliations:** 1Inclusive Medical Science Research Institute, Morinomiya University of Medical Sciences, Osaka 559-8611, Japan; riverwest1004@yahoo.co.jp (K.K.); akihiro_kakimoto@morinomiya-u.ac.jp (A.K.); masahiro_tsutsumi@morinomiya-u.ac.jp (M.T.); isao_yamaguchi@morinomiya-u.ac.jp (I.Y.); 2Department of Rehabilitation, Kano General Hospital, Osaka 531-0041, Japan; 3Department of Radiological Sciences, Faculty of Health Sciences, Morinomiya University of Medical Sciences, Osaka 559-8611, Japan; 4Graduate School of Health Science, Morinomiya University of Medical Sciences, Osaka 559-8611, Japan; 2021mhs001@s.morinomiya-u.ac.jp; 5Department of Physical Therapy, Morinomiya University of Medical Sciences, Osaka 559-8611, Japan; 6AR-Ex Medical Research Center, Tokyo 158-0082, Japan

**Keywords:** neuropathy, tibial nerve tension, deep learning, ultrasound imaging, convolutional neural network

## Abstract

Peripheral nerve tension is known to be related to the pathophysiology of neuropathy; however, assessing this tension is difficult in a clinical setting. In this study, we aimed to develop a deep learning algorithm for the automatic assessment of tibial nerve tension using B-mode ultrasound imaging. To develop the algorithm, we used 204 ultrasound images of the tibial nerve in three positions: the maximum dorsiflexion position and −10° and −20° plantar flexion from maximum dorsiflexion. The images were taken of 68 healthy volunteers who did not have any abnormalities in the lower limbs at the time of testing. The tibial nerve was manually segmented in all images, and 163 cases were automatically extracted as the training dataset using U-Net. Additionally, convolutional neural network (CNN)-based classification was performed to determine each ankle position. The automatic classification was validated using five-fold cross-validation from the testing data composed of 41 data points. The highest mean accuracy (0.92) was achieved using manual segmentation. The mean accuracy of the full auto-classification of the tibial nerve at each ankle position was more than 0.77 using five-fold cross-validation. Thus, the tension of the tibial nerve can be accurately assessed with different dorsiflexion angles using an ultrasound imaging analysis with U-Net and a CNN.

## 1. Introduction

The tibial nerve is a branch of the sciatic nerve that arises at the apex of the popliteal fossa, continuing its course down the leg, posterior to the tibia, and running posteriorly and inferiorly to the medial malleolus through a structure known as the tarsal tunnel [1]. Since it has both motor and sensory functions, compressive neuropathies of the tibial nerve and its branches to the hindfoot can cause heel pain; these can be due to tarsal tunnel syndrome, nerve entrapment, or diabetes-related neuropathy. Some patients with plantar heel pain are diagnosed with plantar fasciitis complicated by tarsal tunnel syndrome [2]. Diabetic peripheral neuropathy, one of the major complications of diabetes mellitus, is difficult to diagnose accurately [3] and can lead to serious complications [4]. Thus, an objective and quantitative assessment method of the condition of the tibial nerve is crucial. 

The nerve conduction study (NCS), which is currently the gold standard method for tibial nerve assessment, can detect neuropathy by detecting the conduction velocity; however, it is time consuming and invasive and easily influenced by skin temperature and humidity [5]. Recently, high-resolution ultrasound (US) has been used to diagnose neuromuscular diseases [6], and shear wave elastography (SWE) has been shown to reflect peripheral nerve stiffness [7,8]. However, SWE for peripheral nerves does not have enough reliability in a clinical setting [7]. Therefore, there is a paucity of precise and reliable diagnostic procedures for the assessment of peripheral nerves using either simple or objective methods.

US imaging devices are ideal for screening in the clinical setting because of their portability, low cost, and noninvasive nature. However, it is difficult to accurately diagnose conditions from US images; diagnostic accuracy depends, to a large extent, on the experience and skill of the examiner. In recent years, advances have been made in the automatic assessment of US images using deep learning [9]; however, there have been very few applications for the neuromuscular system. 

Recent ultrasonographic studies report that changes in tibial nerve morphology, such as the cross-sectional area [10] and nerve stiffness [8], are crucial factors in peripheral neuropathy of the foot. The tension of the tibial nerve is known to increase with increasing ankle dorsiflexion. Thus, developing an automatic assessment system for B-mode US images of the tibial nerve at different ankle dorsiflexion positions can provide the basis for an accurate diagnosis of plantar heel pain and diabetic peripheral neuropathy. However, time is limited in clinical settings, so deep learning assessments of the tibial nerve are considered helpful.

In this context, the aim of this study was to develop an automatic method for assessing B-mode US imaging of the tibial nerve in different ankle positions using deep learning.

## 2. Materials and Methods

### 2.1. Participants

We investigated 68 right ankles of 68 healthy adults (33 men and 35 women; mean age: 20.4 ± 0.7 years; height: 165.1 ± 8.0 cm; weight: 57.7 ± 8.9 kg). A history of orthopedic or neurological disease of the lower limbs or trunk was considered a criterion for exclusion. Ethics approval was granted by the University Ethics Committee (authorization no. 2022-046). Informed consent was obtained from all participants prior to testing. This study was conducted in accordance with the principles of the Declaration of Helsinki.

### 2.2. Ultrasound Image Capturing Method

The tibial nerve was assessed using a B-mode US imaging system (Aplio 300; Canon Medical Systems, Tokyo, Japan) with a 10 MHz linear transducer (PLT-1005BT; Canon Systems, Tokyo, Japan). The imaging area was set based on previous studies [11,12]. The tibial nerve was located using a transverse scan, approximately 1 cm superior to the medial malleolus (Figure 1a), and it was identified using the tibial artery as a landmark (Figure 1b). The nerve was positioned at the center of the screen. The US transducer was rotated by 90° and aligned longitudinally along the tibial nerve plane (Figure 1c). The probe was fixed with a thermoplastic fixture and an elastic bandage. For measurements, the participant was seated on a Biodex 4 isokinetic dynamometer device (Biodex Medical System Inc., New York, NY, USA) and the ankle was fixed to the footplate. The participants were seated in the mid-neck position with 90° flexion of the hip and 30° flexion of the knee. The neck and trunk were rested on the headrest and backrest, respectively. The trunk and right thigh were fixed using a belt, and the participant’s ankle was placed in the maximum passive dorsiflexion position. The tibial nerve was assessed using a B-mode US imaging system by a physical therapist with two years of research experience in US imaging and 10 years of experience in a hospital rehabilitation department.

The motion task consisted of repetitive movements in a range of 20° from the maximum dorsiflexion position of the ankle in the direction of plantar flexion. The ankle was moved at a constant velocity of 30°/s; 0.67 s was allowed for each dorsiflexion and plantar flexion and 0.33 s was allowed to switch the direction of movement. After confirming that the participant was relaxed, the dynamics of the tibial nerve were imaged while moving the ankle from plantar flexion to dorsiflexion in three trials (frame rate, 60; depth, 4.5 cm). 

US videos of the tibial nerve were converted to still images in three ankle positions: at the maximum dorsiflexion position of the ankle and at angles of −10° and −20° from the maximum dorsiflexion of the ankle (Figure 2 and Figure 3). A total of 204 data points were collected for analysis. The Portable Network Graphics (PNG) format was used for the two-dimensional US images with a pixel size of 1536 × 1024. 

### 2.3. Manual Segmentation

The US image analysis procedure is shown in Figure 4. The tibial nerve regions in each US image classifying three different ankle joint angles were extracted for classification and used as U-Net training images. A total of 204 still images (1536 × 1024 pixels) classified as three different ankle joint angles were manually segmented for the tibial nerve using “labelme”, an annotation tool developed by a physical therapist with six years of research experience in US imaging and 15 years of experience in a hospital rehabilitation department.

### 2.4. Convolutional Neural Network (CNN)

There are many deep learning methods for classification and identification. In this study, we chose a CNN, the simplest classification method, to assess the tibial nerve. The learning and test processes were run using a graphic processing unit (NVIDIA GeForce RTX 3080 Ti 12GB of memory), TensorFlow 2.4, and Keras 2.4. The input data were downsized to 384 × 256 pixels to prevent memory overflow. The CNN architecture was constructed as a simple three-layer structure, as shown in Figure 5. This model predicted three classes: maximum dorsiflexion (md) and md −20° and md −10° positions. One layer consisted of two convolutions (kernel size was 3 × 3), a rectified linear unit (ReLU), and a max pooling operation (2 × 2) for downsampling.

Three datasets were used as the input images. Three datasets had 204 data for each of the 68 participants. The first dataset was a raw US image that contained Digital Imaging and Communications in Medicine (DICOM) tag information strings, such as the acquisition conditions and distances. The second dataset was the processed image with strings deleted from the original image. The third dataset was the image obtained by manually extracting only the tibial nerve from the original image. In addition, the pixel size did not change for any image.

### 2.5. U-Net

The U-Net architecture was used to segment the tibial nerve. U-Net is a commonly used technique for the segmentation of biological images. This architecture is composed of encoding and decoding paths, as shown in Figure 6. The process of each layer in the encoding path consisted of a 4 × 4 convolution, a leaky rectified linear unit (LReLU), and a 2 × 2 max pooling operation for downsampling. The process of each layer in the coding path was composed of a 2 × 2 deconvolution, an ReLU, and concatenation with the corresponding cropped feature map from the encoding path for upsampling. The Dice similarity coefficient loss in the output layer was used in this study.

### 2.6. Validation

Five-fold cross-validation was used to evaluate the accuracy of prediction using the CNN learning model. The training and test groups were randomly divided into five sets, as shown in Table 1.

The training and test groups were randomly divided into five sets. In each set, the learning model was calculated using the images of the training group, and the classes of the test group were predicted using the learning model.

In each set, the learning model was calculated using the images of the training group, and the classes of the test group were predicted using the learning model. The accuracy of each set was evaluated for each of the three types of input datasets: raw images, images with DICOM tag information removed, and manually extracted images of the tibial nerve. Finally, the validation of auto-segmentation using U-Net was assessed by the intersection over union (IoU) and cross-sectional area ratio (CSAR). We used both metrics to measure the accuracy of our model by evaluating the area ratio and the alignment of the predicted and ground truth images [13].

## 3. Results

### 3.1. Segmentation

The average accuracies of tibial nerve segmentation using U-Net were verified as 0.81 for IoU and 0.98 for CSAR (Table 2).

The validation of auto-segmentation using U-Net was assessed by the intersection over union and the cross-sectional area ratio.

### 3.2. Classification 

The five-fold cross-validation results showed that the accuracy of the raw data was low (0.44). However, the manual and full auto segmentation accuracies were higher at 0.92 and 0.77, respectively (Table 3). The F values for the raw data could not be calculated as the values diverged. On the other hand, the manual and full auto segmentation F-values were higher at 0.92 and 0.76, respectively (Table 3).

In each set, the learning model was calculated using the images of the training group, and the classes of the test group were predicted using the learning model.

## 4. Discussion

The aim of this study was to develop an automatic assessment of tibial nerve tension using B-mode US imaging with deep learning. We demonstrated a high accuracy in the classification of the tibial nerve for each ankle position (more than 77%) by automatically extracting U-Net and CNN-based classification. 

The applications of deep learning techniques in medical imaging analyses are divided into three types of tasks: “classification”, “detection”, and “segmentation”. There have been few studies on all these types of tasks in US imaging [9].

Regarding segmentation, several prior studies have investigated cardiac and fetal systems [14,15,16]; however, there are very few deep learning studies on the musculoskeletal system. Belasso et al. [17] showed that US images of the lumbar multifidus muscle could be segmented automatically. Nevertheless, few studies have segmented long-axis images of peripheral nerves. 

The classification of images is accomplished by identifying certain anatomical or pathological features that can discriminate one anatomical structure or tissue from others [9]. Studies using this technique have been conducted to diagnose breast tumors [18], liver cancer [19], thyroid nodules [20], and fetal conditions [21].

The results of the present study indicate that automatic segmentation of the tibial nerve and its changes with ankle position can be classified with a high accuracy using deep learning. The tension of the tibial nerve increases with dorsiflexion of the ankle [22,23]; therefore, the US images of the tibial nerve can be classified according to the ankle position, which is equivalent to distinguishing the tension of the tibial nerve.

It is well known that the separation of the target from surrounding structures is difficult in low-contrast US images [9]. Peripheral nerves are surrounded by an epineural membrane that separates them from the surrounding tissues. Therefore, peripheral nerves can be segmented with a high accuracy. Moreover, stretching the tibial nerve during dorsiflexion of the ankle alters the image of the nerve bundles and perineurium within the nerve, since it has the inherent ability of excursion and stretching with the motion of the limb [24]. Therefore, automatic assessments of the tibial nerve in different ankle positions can achieve high accuracy. 

Manual segmentation methods are time consuming but reliable. It is known that the tension of a peripheral nerve can be quantitatively assessed using SWE. US imaging has several advantages compared to other medical imaging modalities, including portability, accessibility, and cost-effectiveness; in addition, only high-end models are available for SWE. Therefore, an automatic assessment system may be useful in the clinical screening of peripheral nerve neuropathy.

This study has some limitations which should be mentioned. First, different US devices may have lower accuracies. Second, only young, healthy volunteers were included in this study. Therefore, it is necessary to perform the same investigation in older adults and individuals with various neuropathies and other conditions.

## 5. Conclusions

The automatic classification was validated using five-fold cross-validation from the testing data composed of 41 data points. The highest mean accuracy (0.92) was achieved using manual segmentation. The mean accuracy of the full auto-classification from the tibial nerve at each ankle position was more than 0.77 using five-fold cross-validation. The tension of the tibial nerve can be accurately assessed with different dorsiflexion angles using US imaging analyses with U-Net and a CNN.

## Figures and Tables

**Figure 1 sensors-23-04855-f001:**
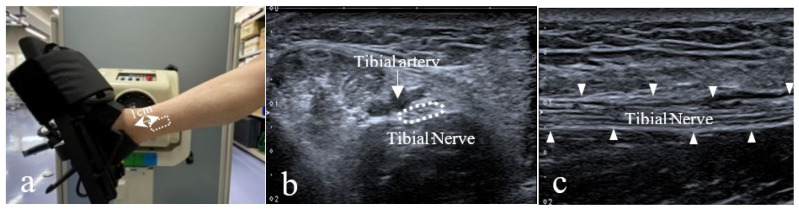
B-mode ultrasound images of the captured location and the tibial nerve. (**a**) Probe location. (**b**) Short-axis image of the tibial nerve. (**c**) Long-axis image of the tibial nerve.

**Figure 2 sensors-23-04855-f002:**
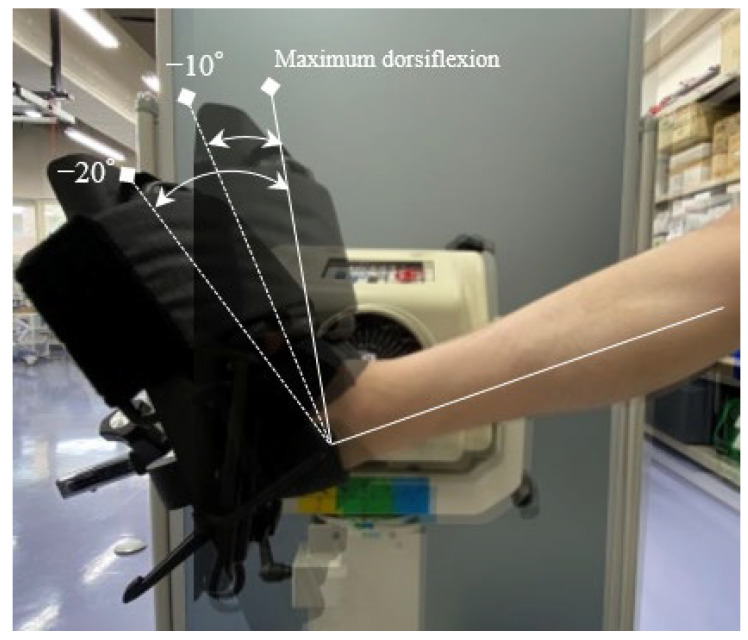
Three ankle positions: at the maximum dorsiflexion position of the ankle and at angles of −10° and −20° from the maximum dorsiflexion of the ankle.

**Figure 3 sensors-23-04855-f003:**
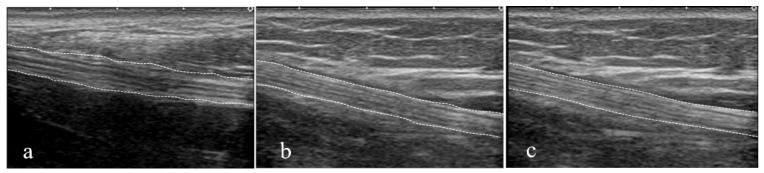
Morphology of the tibial nerve in three different ankle joint angles. (**a**) −20° from maximum dorsiflexion. (**b**) −10° from maximum dorsiflexion. (**c**) Maximum dorsiflexion.

**Figure 4 sensors-23-04855-f004:**
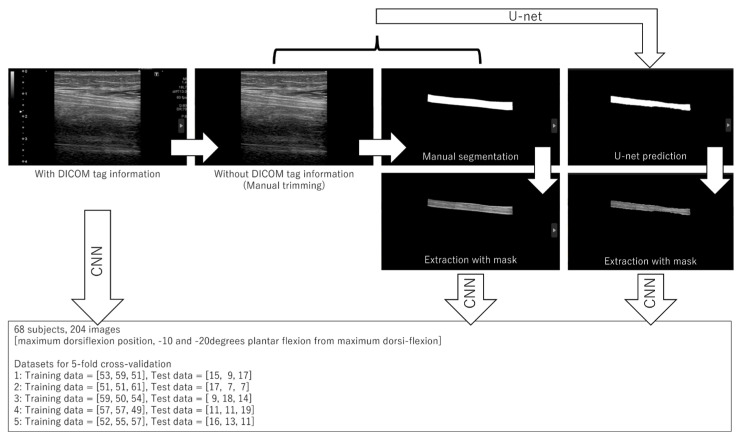
The US image analysis procedure.

**Figure 5 sensors-23-04855-f005:**
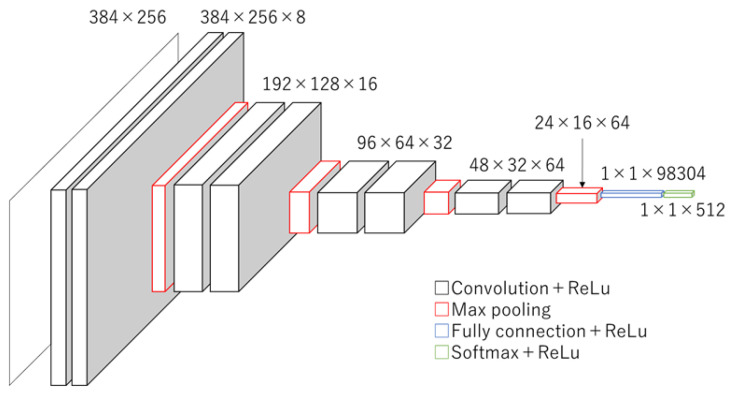
Structure of the CNN. The CNN architecture was constructed as a simple three-layer structure.

**Figure 6 sensors-23-04855-f006:**
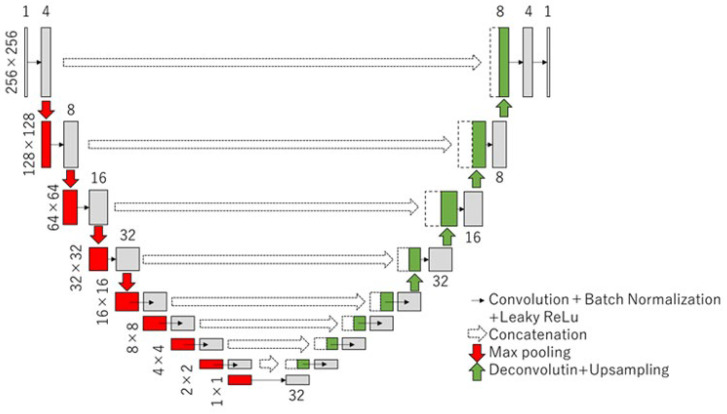
Architecture of the U-Net. This architecture is composed of encoding and decoding paths.

**Table 1 sensors-23-04855-t001:** Classification for five-fold cross validation.

Class	Set 1		Set 2		Set 3		Set 4		Set 5	
	Training	Test	Training	Test	Training	Test	Training	Test	Training	Test
Md −20	53	15	51	17	59	9	57	11	52	16
Md −10	59	9	51	17	50	18	57	11	55	13
Md	51	17	61	7	54	14	49	19	57	11

**Table 2 sensors-23-04855-t002:** Accuracy of tibial nerve segmentation using U-Net.

	Intersection over Union	Cross-Sectional Area Ratio
1	0.81	1
2	0.79	0.99
3	0.80	0.94
4	0.81	1
5	0.82	0.98
Accuracy Average	0.81	0.98

**Table 3 sensors-23-04855-t003:** Results of five-fold cross-validation.

	Raw Data		Manual Segmentation		Full Auto Segmentation	
	Accuracy	F-Value	Accuracy	F-Value	Accuracy	F-Value
1	0.37	-	0.98	0.98	0.83	0.82
2	0.59	-	0.95	0.95	0.80	0.79
3	0.22	-	0.83	0.83	0.66	0.67
4	0.27	-	0.93	0.93	0.80	0.81
5	0.73	-	0.93	0.93	0.75	0.73
Average	0.44	-	0.92	0.92	0.77	0.76

F values for the raw data could not calculated as the values diverged.

## Data Availability

The datasets generated and/or analyzed during the current study are available from the corresponding author upon reasonable request.

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
