# Peer review of "Automatic Identification of Ultrasound Images of the Tibial Nerve in Different Ankle Positions Using Deep Learning"

_sensors, 2023, doi:10.3390/s23104855_

Round 1
Reviewer 2 Report
This manuscript is an image-based AI analysis, this manuscript was well-prepared. However, there are also some concerns need to be addressed.
1. In the introduction section, the authors should introduce the significance of this study.
2. The figure legend is too simple, I recommend the authors supply more detailed information in the figure legend.
The expression is fine.
Reviewer 3 Report
The authors reported a deep learning algorithm for the automatic assessment of tibial nerve tension using B-mode 19 ultrasound imaging. The highest mean accuracy (0.92) was achieved using manual segmentation. The tension of the tibial nerve can be accurately assessed with different dorsiflexion angles using US imaging analysis with U-Net and CNN. The data is well organized and solid, the discussion is clear and the conclusion is sound. I recommend publication of this work in Sensors.
Author Response
May 15, 2023
The Editorial Board
Sensors
Dear Editor:
RE: Submission of a revised manuscript (ID: sensors-2396073)
Dear Editors:
Thank you for the helpful comments and important insights provided by you and the reviewers. We are grateful for the time and effort dedicated to reviewing our manuscript.
Thank you for your consideration. I look forward to hearing from you.
Sincerely,
Shintarou Kudo, PT, PhD
Graduate School of Health Science
Morinomiya University of Medical Science
1-26-16 Nankoukita Suminoe Ward
Osaka City, Osaka Prefecture 559-8611, Japan
Telephone: +81-6-6616-6911
Fax: +81-6-6616-6912
Email: kudo@morinomiya-u.ac.jp
Reviewer 4 Report
Dear authors,
I want to congratulate the authors for their work. The manuscript is well-written and concise.
Also, I have some suggestions in order to improve the paper:
- the same operator performed all ultrasounds, or were there different operators ( different experience)?
-in section 2 , you could add another subsection : Participants/ Patients selection. I suggest you to add the inclusion/ exclusion criteria
-results: Add a brief description of your cohort group ( mean age, gender , background )
-maybe Future directives/ perspectives subchapter could be interesting
I look forward to receiving the modified version.
English is fine. Minor corrections are required.
